# A Deep Batch Normalized Convolution Approach for Improving COVID-19 Detection from Chest X-ray Images

**DOI:** 10.3390/pathogens12010017

**Published:** 2022-12-22

**Authors:** Ibrahim Al-Shourbaji, Pramod H. Kachare, Laith Abualigah, Mohammed E. Abdelhag, Bushra Elnaim, Ahmed M. Anter, Amir H. Gandomi

**Affiliations:** 1Department of Computer Science, University of Hertfordshire, Hatfield AL10 9AB, UK; 2Department of Electronics & Telecommunication Engineering, Ramrao Adik Institute of Technology, Nerul, Navi Mumbai 400706, Maharashtra, India; 3Computer Science Department, Prince Hussein Bin Abdullah Faculty for Information Technology, Al al-Bayt University, Mafraq 25113, Jordan; 4Hourani Center for Applied Scientific Research, Al-Ahliyya Amman University, Amman 19328, Jordan; 5Faculty of Information Technology, Middle East University, Amman 11831, Jordan; 6Applied Science Research Center, Applied Science Private University, Amman 11931, Jordan; 7School of Computer Sciences, Universiti Sains Malaysia, Pulau Pinang 11800, Malaysia; 8Department of Information Technology and Security, Jazan University, Jazan 45142, Saudi Arabia; 9Department of Computer Science, College of Science and Humanities in Al-Sulail, Prince Sattam bin Abdulaziz University, Riyadh 11671, Saudi Arabia; 10Egypt-Japan University of Science and Technology (E-JUST), Alexandria 21934, Egypt; 11Faculty of Computers and Artificial Intelligence, Beni-Suef University, Benisuef 62511, Egypt; 12Faculty of Engineering and IT, University of Technology Sydney, Ultimo, NSW 2007, Australia; 13University Research and Innovation Center (EKIK), Óbuda University, 1034 Budapest, Hungary

**Keywords:** chest X-ray, COVID-19, deep learning, batch normalized convolutional neural network (BNCNN), classification

## Abstract

Pre-trained machine learning models have recently been widely used to detect COVID-19 automatically from X-ray images. Although these models can selectively retrain their layers for the desired task, the output remains biased due to the massive number of pre-trained weights and parameters. This paper proposes a novel batch normalized convolutional neural network (BNCNN) model to identify COVID-19 cases from chest X-ray images in binary and multi-class frameworks with a dual aim to extract salient features that improve model performance over pre-trained image analysis networks while reducing computational complexity. The BNCNN model has three phases: Data pre-processing to normalize and resize X-ray images, Feature extraction to generate feature maps, and Classification to predict labels based on the feature maps. Feature extraction uses four repetitions of a block comprising a convolution layer to learn suitable kernel weights for the features map, a batch normalization layer to solve the internal covariance shift of feature maps, and a max-pooling layer to find the highest-level patterns by increasing the convolution span. The classifier section uses two repetitions of a block comprising a dense layer to learn complex feature maps, a batch normalization layer to standardize internal feature maps, and a dropout layer to avoid overfitting while aiding the model generalization. Comparative analysis shows that when applied to an open-access dataset, the proposed BNCNN model performs better than four other comparative pre-trained models for three-way and two-way class datasets. Moreover, the BNCNN requires fewer parameters than the pre-trained models, suggesting better deployment suitability on low-resource devices.

## 1. Introduction 

The coronavirus (COVID-19) remains a global health problem that negatively impacts our lives and the global economy. The initial infection was reported on January 2020 of twenty-seven patients with Pneumonia and an epidemiological link to a live wild animal market [1]. The actual host remained unknown; however, the transmission can be through inhaling duplets produced by the cough or sneeze of an infected person within six feet. Likewise, studies have shown that the transmission can be through touching virus-covered surfaces and passing it to their mouth, nose, or possibly eyes [2,3]. To tackle this, there has to be a solution where the suspected patients receive faster confirmation of the presence of the disease. This can be provided by imaging modalities, such as chest radiographs (X-ray) or Computed Tomography (CT) scan images. The radiologists can give an opinion by analyzing and applying some image analysis methods to diagnose COVID-19 cases. It was confirmed that X-rays or CTs contain valuable information related to this disease. Therefore, a faster and more accurate diagnosis can be performed using radiologic images with Machine Learning (ML) techniques to determine COVID-19 cases [4]. 

Nowadays, ML techniques have become essential tools in medical diagnosis, and several studies have confirmed their capabilities for diagnosing COVID-19 [2,3,5,6]. Furthermore, these techniques assist physicians and health experts in diagnosing patients with pulmonary embolism, pulmonary circulation, or body temperature to improve the speed and accuracy in detecting COVID-19 cases [7,8,9,10]. Recently, the deep learning algorithm has gained special attention due to its potential to effectively extract relevant features from medical images as a part of its search process and classify them for disease diagnosis. These models can help radiologists and health experts to interpret, identify, and triage positive-infected COVID-19 cases.

Visual Geometry Group (VGG)-16, VGG-19, and other pre-trained models require large-scale datasets to optimize their model parameters in TL [11]. Although Transfer Learning (TL) models allow flexibility in retraining some or all the layers of such models for the desired task, the output of most layers remains biased due to a considerable number of pre-trained weights and parameters by these models [12]. Despite promising results reported for COVID-19 detection from chest X-ray images, most existing approaches have not provided parameter requirements in the pre-trained models to improve model classification capability. Therefore, further efforts are still needed to reduce computation complexity and the required parameters to train those models. In this paper, a novel model, namely BNCNN, is proposed. It uses batch normalization [13] to increase the model’s generalization and dropout layers [14] to improve training speed and avoid the model overfitting COVID-19 detection from X-ray images. The BNCNN can be considered a diagnostic system for COVID-19, which is necessary for critical conditions, as it can be helpful to the radiologists and their decisions in specifying COVID-19 cases from X-ray images. The main contributions of this paper can be summarised as follows:

A BNCNN model comprising a pre-processor, a feature extractor, and a classifier is proposed. The pre-processor resizes and normalizes input X-ray images to improve feature mapping and reduce classification loss. The feature extractor comprises four repetitions of the cascaded convolution layer to learn feature maps, a batch normalization layer to solve the internal covariance shift of feature maps, and a max-pooling layer to increase the convolution span for generating high-level patterns. The classifier comprises two repetitions of a dense layer to learn complex feature maps, a batch normalization layer to standardize internal feature maps, and a dropout layer to avoid overfitting while improving generalization. The proposed BNCNN model retains the pre-processor and features extractor while only the number of neurons in the output layer of the classifier is changed for binary and muti-class COVID-19 detection. 

The performance of the BNCNN model is investigated against four pre-trained models, namely, VGG-16, VGG-19, Inception-V3, and ResNet-50, for COVID-19 detection from X-ray images. 

The proposed model requires fewer parameters than other pre-trained models making it more suitable for implementation on low-resource devices. 

This paper is organized as follows: Section 2 describes the COVID-19 dataset, feature extraction, and adaptation of the pre-trained image classification models, and Section 3 presents experimental results. Finally, Section 4 concludes the paper.

## 2. Proposed Model 

The pre-trained VGG models inspire the proposed model intending to reduce computation complexity while increasing the classification accuracy of COVID-19 detection. Various tasks explored the state-of-the-art VGG models due to their excellent capability for feature extraction. These models can be well understood in two sections: Feature extraction and classifier. The first section embeds the raw input into low-dimensional vectors further accepted by the classifier for generating desired class labels. The proposed BNCNN is inspired by the repetition of block structures of feature extraction, as in the VGG model. The proposed BNCNN-based COVID-19 detection system has three primary phases: Data pre-processing, Feature extraction, and Classification. We explain each phase in the following subsections.

### 2.1. Data Pre-Processing

The proposed model uses chest X-ray images of COVID-19 patients and other subjects. Researchers from the University of Doha and the University of Dhaka collected this dataset, and it is publicly available with metadata on the ‘Kaggle’ [15]. The dataset contains three classes, including COVID-19, Normal and Viral Pneumonia images. Samples of X-ray images from each of the three classes in the dataset are provided in Figure 1. For experiments and evaluation, the dataset is partitioned into three mutually exclusive and exhaustive subsets for training (80%), validation (10%), and testing (10%). These subsets are summarised in Table 1.

The X-ray images in the dataset are uniformly pre-processed to facilitate the learning process. Each X-ray image is resized to 150 × 150 pixels, unlike 224 × 224 as in VGG models. It reduces the network’s input dimension and trainable and non-trainable weights of the BNCNN model. Each X-ray image is divided by 255, resulting in a normalized image in 0–1, which facilitates weight learning by avoiding vanishing and exploding gradients. Data augmentation strategies are used to simulate real-life scenarios and avoid the risks of overfitting. All subsets are augmented independently with a rotation ranging from −10 to 10 degrees, a zooming range of 0–10%, shearing of 0–10%, a horizontal stride of 0–10%, a vertical stride of 0–10%, and horizontally flip to improve generalization and increase diversity in the learning process by the models. The pixel values unavailable in the augmented image are replaced by the nearest pixel values. Examples of pre-processed chest X-ray images from each of the three classes are shown in Figure 1. The vertical flip is avoided because it will be easy for an ordinary user to identify vertical orientation in chest X-ray images. The class label for each image is encoded using one-hot encoding. It converts the label into a 3D vector with all zeros except for the one corresponding to the image class. It can be noted that the actual order of the classes in these three dimensions does not affect the classifier performance [16].

### 2.2. Feature Extraction

The feature extraction phase comprises the first twelve layers, while the remaining are for the classifier phase. Architectural details of the BNCNN model, the output dimension at each layer, and the number of trainable/non-trainable parameters are provided in Table 2. Inspired by the VGG models, the feature extraction phase of the proposed BNCNN uses four repetitions of a block of similar layers. Each block comprises a convolution layer to learn kernel weights suitable for the features map, followed by a batch normalization layer to solve the internal covariance shift of feature maps [13] and a max-pooling layer to find the highest-level patterns in the input images by increasing the span of the convolution calculation [16].
(1)$$f_{max}\left(m,n\right)=max\left(I\left(h,\;v\right),\;I\left(h+1,\;v\right),\;I\left(h,\;v+1\right),\;I\left(h+1,\;v+1\right)\right)$$
where h=m+s and v=n+s.

Herein, m and n are horizontal and vertical indices of the image, and s is stride. A detailed illustration can be found in [17]. Each convolution layer comprises 3 × 3 filters with stride and padding of 1 and a ReLU activation function. The number of filters in convolution layers increases from the input to the output layers. A batch normalization layer after every convolution layer increases the model’s generalization capability. It standardizes the output of the previous layer to have a mean of zero and a Standard Deviation of one. These layers keep track of input variable statistics during training and standardize input during testing. The standardized variables can be scaled at the transformed output to have the desired statistics updated during training and maintained for testing. Hence, these layers have an equal number of trainable and non-trainable parameters. A max-pooling layer reduces the dimension of the input feature map without any parameters. All max-pooling layers use a stride and maximum size of two, halving the dimension of the input feature map at each occurrence.

### 2.3. Classification

The classifier starts with the flattening layer to convert all output of the previous phase (i.e., feature extraction) into a vector. This phase replaces convolution and max-pooling layers with dense and dropout layers. It comprises two repetitions of a block of the dense layer for feature mapping, followed by a batch normalization layer to standardize internal feature maps. The dropout layer avoids overfitting while aiding the model generalization, and a softmax layer generates a probabilistic output for each class label. 

The number of neurons in the dense layer decreases from 256 to 128 from input to output to remove redundant features. Furthermore, it reduces the model’s computational complexity as the number of trainable parameters for dense layers increases exponentially with the number of neurons. All dropout layers use a drop factor of 0.2, indicating that randomly selected 20% weights are updated in each iteration to increase the model’s generalization. 

The architecture of the BNCNN remains unchanged for three-way and two-way classifications except for the softmax layer. In three-way classification, the softmax layer uses three output nodes with 387 trainable parameters, resulting in 2,786,435 trainable parameters. While in two-way classification, two output nodes with 258 trainable parameters are used, resulting in 2,786,306 trainable parameters. 

The Adaptive Moment Estimation (Adam) optimizer iteratively updates network weights using the training data. This optimizer combines Stochastic Gradient Descent (SGD) and Root Mean Square Propagation (RMSP). The choice of Adam optimizer facilitates adjusting the learning rate for each weight in the network by computing the first and second moments of the gradient, adaptive learning rate, and history-based updates for faster convergence. Moreover, it shows the best accuracy compared to SGD and RMSP optimization algorithms [17].

For the *n*th training example, let $y_{pred}$(*n*, *i*) be the *i*th scalar value corresponding to predicted class probability ($\sum_{i}y_{pred}\left(n,\;i\right)=1,\;\forall n$) and *y* (*n*, *i*) be the *i*th scalar value corresponding to the actual one-hot-encoded class label. The cross-entropy loss (*L*) is calculated as follows:(2)$$L=-\overset{N_{train}}{\underset{n=1}{\sum }}\sum_{i=1}^{N_{class}}y\left(n,\;i\right)\;.\;\log y_{pred}\left(n,\;i\right)$$
where, $N_{train}$ is the total number of training examples and $N_{class}$ is the number of classes (i.e., 3 for three-way and 2 for two-way classification). 

The number of non-trainable parameters remains unaltered in both classification types. The major contributors to the increasing number of trainable parameters, convolution, and dense layers are minimized in the proposed BNCNN layer. Figure 2 illustrates the overall process of the BNCNN model.

A callback is tailored to reduce the learning rate after every epoch if the validation accuracy stops improving. The callback’s primary goal is to monitor the validation accuracy and reduce the learning rate by a factor of 0.3 if no improvement is achieved for three consecutive epochs. After reducing the learning rate, it waits for at least five epochs before applying the reduction again. Another callback is tailored for early stopping the training by monitoring training and validation accuracies after each epoch. The callback stores the model weights if the current training and validation accuracies exceed the earlier epochs. The second callback avoids model overfitting without worrying about the exact number of epochs. The proposed BNCNN steps are represented in Algorithm 1.
**Algorithm 1:** BNCNN model***INPUT****IMG*: Dataset of X-ray images [224 × 224 × 15,153]*LAB*: Set of labels {‘COVID-19’, ‘Normal’, ‘Viral Pneumonia’} corresponding to X-ray images [1 × 15,153]Nclass: number of classes used for training***OUTPUT***Ypred: a matrix of prediction probability of class labels [Nclass × 683]***ALGORITHM***:If 3-way Nclass  = 3If 2-way Nclass = 2For *I* = 1 to length (*LAB*)If *LAB* (1, *i*) == ‘COVID-19’ or *LAB*(1, *i*) == ‘Normal’*X*(:, :, *i*), *Y*(1, *i*) = *IMG*(:, :, *i*), *LAB*(1, *i*)End ifEnd_forElseNclass = 2*X*, *Y* = *IMG, LAB*End_if*X* = image_resize(*X*)/255Yenc = one-hot_encoder (*Y,*
Nclass)If *model_name* == ‘VGG-16’ or *model_name* == ‘VGG-16’*model* = **load_pre-trained_weigts**(*model_name*)*model* = freeze_feature_extration_layers(*model*)Else_if *model_name* == ‘BNCNN’ *model* = **construct_model**() End_ifXtrain, Xval, Xtest, Ytrain, Yval, Ytest
*= **cross-validation** (X*, Yenc, *train =* 0.8, *val =* 0.1, *test =* 0.1*)*Xtrain_aug, Ytrain_aug
*= **data_augmentation**(*Xtrain, Ytrain, *rotation =* [−20, 20], *zoom =* [0, 0.2], *shear =* [0, 0.2], *horizontal_flip =* True, *vertical_flip =* False*)*Xval_aug, Yval_aug
*= **data_augmentation**(*Xtrain, Ytrain, *rotation =* [−20, 20], *zoom =* [0, 0.2], *Shear =* [0, 0.2], *horizontal_flip =* True, *vertical_flip =* False*)**model = **train** (*Xtrain_aug, Ytrain_aug, Xval_aug, Yval_aug, *custom_callbacks)*Ypred
*= **predict** (model*, Xtest*)**cm = **calculate_confusion_matrix**(*Ypred, Ytest, Nclass*)**acc*, *recall*, *precision*, *F1-score = **calculate_evaluation_metrics**(cm*, Nclass*)*

## 3. Experiments and Results

This section provides the implementation details of the BNCNN model, followed by the results of the experiments. 

### 3.1. Experimental Setup 

The BNCNN model and other pre-trained models: VGG-16, [18], VGG-19, [18], Inception-V3, [19] and ResNet-50, [20] are evaluated on the dataset to classify chest X-ray images into 3-way classification: COVID-19, Normal and Viral Pneumonia and for two-way classification: COVID-19 and Normal X-ray images. 

The BNCNN model is trained for 100 iterations using the Adam optimizer with an initial learning rate decay of 0.0001 to finish all the epochs and obtain the solutions without interruption. The cross-entropy loss function minimizes the distance between predicted and actual probability distributions. 

The hyper-parameter settings of the Adam optimizer for the BNCNN model and other pre-trained models are provided in Table 3. These settings are assigned after we experimentally find that these are the best settings of parameters for training the models. All the models are implemented using Python and are executed using 12 GB NVIDIA Tesla P100 GPU and Intel Xenon CPU @ 2.00GHz with 13 GB RAM.

### 3.2. Evaluation Measures 

For model evaluation, accuracy, sensitivity, Positive Predictive Value (PPV), and F1-score measures are used to assess the performance of the BNCNN and the other models. The equations for deriving the values of these metrics are provided in (3)–(6).
(3)$$\;\mathrm{Accuracy}\;\left(\mathrm{Acc}\right)=\frac{\mathrm{TP}+\mathrm{TN}}{\mathrm{TP}+\mathrm{TN}+\mathrm{FN}+\mathrm{FP}}$$
(4)$$\mathrm{Sensitivity}\;\left(\mathrm{Sen}\right)=\frac{\mathrm{TP}}{\mathrm{TP}+\mathrm{FN}}$$
(5)$$\mathrm{Positive}\;\mathrm{Predictive}\;\mathrm{Value}\;\left(\mathrm{PPV}\right)=\frac{\mathrm{TP}}{\mathrm{TP}+\mathrm{FP}}$$
(6)$$F1-\mathrm{score}\;\left(F1\right)=2\;\frac{\mathrm{Percision}\;\times \mathrm{Recall}}{\mathrm{Percision}+\mathrm{Recall}}$$

True Positive (TP) denotes the cases where the predicted class label is the same as the class under consideration. True Negative (TN) refers to the cases where classes not under consideration are predicted as themselves. False Negative (FN) is a misclassified case where the class under consideration is predicted as any class other than itself. False Positive (FP) indicates the cases where the model wrongly identifies other classes as the class under consideration. 

## 4. Results and Discussion

This section discusses the comparative performance of the BNCNN against the other pre-trained models.

### 4.1. Results of the Proposed BNCNN Model 

The comparative performance results of the BNCNN model and the pre-trained models for three-way classification are provided in Table 4. Although the VGG-16 model performs slightly better than the proposed BNCNN model during training, the latter performs better during the validation and testing phases, as shown in Table 4. The ResNet-50 achieved the least performance results in training, validation, and testing datasets, followed by VGG-19 and Inception-V3. This is due to the large number of pre-trained network parameters causing excessive bias in the feature extraction phase. For three-way COVID-19 classification, the BNCNN model achieved an accuracy of 96.84% (95% CI: 91.26–97.45), a sensitivity of 93.06% (95% CI: 89.13–96.54), PPV of 97.40% (95% CI: 94.71–98.80), and F1 of 95.18% (95% CI: 88.45–0.97.13) in the testing phase. 

For two-way classification, the performance results of the BNCNN model are comparatively higher than the VGG-16, VGG-19, Inception-V3, and ResNet-50, as shown in Table 5. The BNCNN model achieved 97.28% accuracy (95% CI: 95.25–98.73), 97.28% sensitivity (95% CI: 95.32–98.75), 97.23% PPV (95% CI: 95.19–98.68) and 97.25% F1 (95% CI: 95.33–98.13) in the training phase. In the validation phase, the BNCNN model achieved 98.55% accuracy (95% CI: 97.45–99.26), 98.52% sensitivity (95% CI: 96.84–99.22), 98.53% PPV (95% CI: 97.03–99.18), and 98.52% F1 (95% CI: 96.95–99.43), while achieving an accuracy, sensitivity, PPV, and F1 in the testing phase are 99.27% (95% CI: 98.35–99.63), 99.45% (95% CI: 98.37–99.73), 98.83% (95% CI: 97.64–99.36), and 99.14% (95% CI: 98.24–99.61), respectively. A similar degradation in the performance of the VGG-19 model can be observed in the three-way classification.

A confusion matrix is utilized to determine further the distribution of the predicted X-ray images in different classes. The confusion matrices for the three-way and two-way classification of the BNCNN model are shown in Figure 3. The BNCNN model is tested on the test dataset, including 384 COVID-19 X-ray images, 981 Normal images, and 153 Viral Pneumonia images for three-way classification. For two-way classification, 369 COVID-19 images and 1004 normal images are used. 

Figure 3a shows that 360 of 384 COVID-19 images, 979 of 981 normal images, and 131 of 153 Viral Pneumonia images are correctly classified for the three-way classification. Figure 3b shows 361 out of 369 COVID-19 images, and 1002 out of 1004 Normal images are classified for two-way classification. 

### 4.2. Convergence Analysis

The convergence analysis is performed to study the stability of the learning patterns by the BNCNN model over the number of epochs. Figure 4 plots the BNCNN model accuracy and loss on the training and validation datasets over the training number of epochs. Figure 4a is for three-way classification, while Figure 4b is for two-way classification. It can be observed from Figure 4 that the BNCNN model has shown the best fit and convergence for three-way and two-way classification.

### 4.3. Comparison with Existing Models 

Several ML models have been developed to diagnose COVID-19 automatically from X-ray images. Ozturk et al. (2020) [21] suggested the DarkCovidNet model for automatic COVID-19 detection from X-ray images. The model used 17 convolution layers with different numbers of filters in each layer. The DarkCovidNet model reported high reliability with an accuracy of 98% for two-way classification (i.e., COVID-19 and Normal) and 87% for 3-way classification (i.e., COVID-19, Normal, and Viral Pneumonia). In another work, Khan et al. (2020) [22] reported a CoroNet model based on pre-trained Inception architecture for COVID-19 detection from X-ray images. The CoroNet reported an accuracy of 95% for three-way classification. 

Apostolopoulos and Mpesiana (2020) [23] suggested a novel convolution neural network (CNN) architecture and examined VGG-19 for COVID-19 detection from X-ray images. The model reported an accuracy of 93.48% for three-way classification. Wang et al. (2020) [24] reported a COVID-Net model for 3-way classification with an accuracy of 92.4%. Sethy and Behera (2020) [25] combined TL based on ResNet-50 and the support vector machine to diagnose COVID-19 from X-ray images. Their combined model achieved 95.38% accuracy for three-way class classification.

Horry et al. (2020) [26] used a pre-trained VGG-19 model for COVID-19 detection using a dataset comprising 115 COVID-19, 60361 Normal, and 322 Pneumonia X-ray images. The results showed that the pre-trained model attained 81% accuracy for 3-way classification. In another work, Rahimzadeh and Attar (2020) [27] detected COVID-19 from X-ray images using several deep neural networks and reported an accuracy of 91.4% for 3-way classification. Song et al. (2021) [28] developed a computer-aided method to classify images into COVID-19, bacterial Pneumonia, and Normal cases from a dataset collected from two provinces in China. Experimental results showed that the reported model could accurately identify COVID-19 cases with an accuracy of 86%. 

Hussain et al. (2021) [29] suggested a CoroDet model for COVID-19 detection from X-ray images. The results confirmed that the CoroDet model could effectively identify COVID-19 cases with an accuracy of 94.2% for three-way and 99.1% for two-way classification. Chen (2021) [30] employed a CNN model to detect COVID-19 cases from X-ray images, and the results showed an accuracy of 85% for 3-way classification. Vinod et al. (2021) [31] suggested DeepCovix-net to effectively diagnose COVID-19 from X-ray and CT medical images and reported an accuracy of 96.8% for 3-way class classification. Anter et al. (2021) [32] proposed a model for COVID-19 diagnosis from X-ray images called AFCM-LSMA. Their suggested model achieved an accuracy of 96% for two-way classification. Basha et al. (2021) [33] reported a neurotrophic model for COVID-19 diagnosis from chest X-ray images with an accuracy of 98.7% for two-way classification.

Table 6 depicts the accuracy of the achievements of the existing models. Most studies in Table 6 used different datasets to validate their proposed model’s efficiency. The dataset used in this work is collected from existing studies and is publicly available [34]. It is not fair to compare the performance of the BNCNN with the other models since the size and chrematistics of the datasets are different, but the performance of these models is still comparable. However, Table 6 depicts the achievements of 13 existing models against the proposed BNCNN model in terms of accuracy for three-way and two-way class classification. As per the results in Table 6, the proposed BNCNN model provides higher accuracy for three-way and two-way classification than the existing models. 

### 4.4. Statistical Analysis

To further evaluate and show the significance of the results of the BNCNN, Friedman’s test is performed [35]. We partially trained models to analyze learning speed based on testing accuracy. Testing accuracy performance for the BNCNN and the other pre-trained models, corresponding to the respective model’s test accuracy, are evaluated after 100 epochs. 

Friedman’s test is performed with the null hypotheses where the testing accuracy samples for BNCNN and other pre-trained models originated from the same distribution, i.e., all models under comparison have equal testing accuracy. The alternate hypothesis assumes at least one of the models predicts different testing accuracy than other COVID-19 detection models with a significance level (*p* < 0.05). The samples for testing accuracy are taken from partially trained models at every ten epochs. It should be noted that higher rankings indicate improved performance. The average ranks of all the models for three-way and two-way classification are shown in Figure 5. The p-value calculated using Friedman’s test for three-way classification is 0.0146 and for two-way classification is *p* = 0.0053, which is less than the value of *p*. The test indicates that testing accuracy is different for comparative models. From Figure 5, BNCNN is ranked first for both three-way and two-way classification.

Holm’s posthoc test is used to confirm the differences in the behavior of the BNCNN (controlled model) and the other comparative models, as provided in Table 7. It uses BNCNN as the controlled model because of the highest rank in Friedman’s test. The results in Table 7 show a significant difference between the BNCNN and other pre-trained models. This proves the efficiency of the BNCNN as an alternative model for COVID-19 detection.

In addition, the ability of the BNCNN and other models are assessed by comparing them in terms of Area under the Receiver Operating Curve (AUC). We used the DeLong test to assess differences between the AUC of the models (*p* ≤ 0.05 is considered statistically significant) [36]. Figure 6 shows the discriminative ability of all the used models.

The proposed BNCNN model showed better discriminative ability with an equal AUC of 0.92 (95% CI: 0.87–0.95) and 0.94 (95% CI: 0.89–0.96) for three-way and two-way classification, respectively. Discriminative analysis of other models compared to the proposed BNCNN is shown in Table 8. No competing model has a statistically similar AUC to the proposed BNCNN model. It can be observed that all models have smaller AUCs than the BNCNN model, indicating the better discriminative ability of the developed BNCNN model.

## 5. Discussion

This work aimed to introduce a novel model for COVID-19 detection from X-ray medical images as an intelligent platform, which can provide updates on the patient’s health conditions and then guide further treatment. The proposed BNCNN model performance and efficacy are investigated using several evaluation measurements and compared to VGG-16, VGG-19, Inception-V3, and ResNet-50 pre-trained models. It is observed from the performance measures comparison that the suggested model shows superior results compared to the tested pre-trained models. In addition, the statistical tests of significance proved the superiority of the proposed model compared to other pre-trained models, which reflects the reliability of the developed BNCNN model. This is because the batch normalization layers in the proposed BNCNN model extracted features much better, while the max pooling and dropout layers reduced the computational complexity in their structure.

The main limitation associated with our study is the dataset, which is restricted only to publically available chest X-ray images. X-ray images are not recommended as the first line-imaging test for diagnosing COVID-19 due to the low positive detection rate at the early stages, which may be related to the insensitivity of X-ray images to the density of Ground Glass Opacity (GGO) [37]. Conversely, CT showed significant advantages in monitoring disease progression and served as an effective clinical diagnostic tool for early screening and diagnosis of COVID-19 [38]. CT has proved to be a good choice for early detection, severity assessment, and timely therapeutic effects evaluation for COVID-19, with or without laboratory confirmation [39]. X-ray and CT are medical imaging techniques widely used to assess and diagnose COVID-19 pneumonia patients. However, CT shows greater sensitivity for early pneumonic change, disease progression, and alternative diagnosis than X-ray [40]. ML can analyze irregular symptoms and other ‘red flags ‘of the infected cases at the early stage by using advanced algorithms [41]. These methods show a promising way for optimizing healthcare and improving the results of diagnostic and therapeutic procedures. Therefore, the extension of the proposed BNCNN model can be improved by using a large CT dataset to build an intelligent, accurate, and cost-effective platform for COVID-19.

## 6. Conclusions and Future Work

The COVID-19 disease is becoming increasingly significant as infected cases rapidly increase. Many researchers have devoted their efforts to developing ML models for COVID-19 detection that would benefit radiologists and health experts. This paper proposed an improved model named BNCNN to detect COVID-19 from chest X-ray images. The BNCNN uses VGG-inspired repetitive block structure, and each block comprises a convolution, followed by batch normalization, and max-pooling layers to improve the model’s generalization and feature map reduction. For confirming the reliability of the proposed BNCNN and other pre-trained models: VGG-16, VGG-19, Inception-V3, and ResNet-50 pre-trained models, a dataset is employed from ‘Kaggle’. The results show the superiority of the proposed BNCNN model over the pre-trained models, which significantly outperforms other comparative pre-trained models. Hence, the proposed BNCNN can be used to recognize the COVID-19 virus accurately. In the future, the performance of the proposed BNCNN model on CT imaging needs to be investigated.

## Figures and Tables

**Figure 1 pathogens-12-00017-f001:**
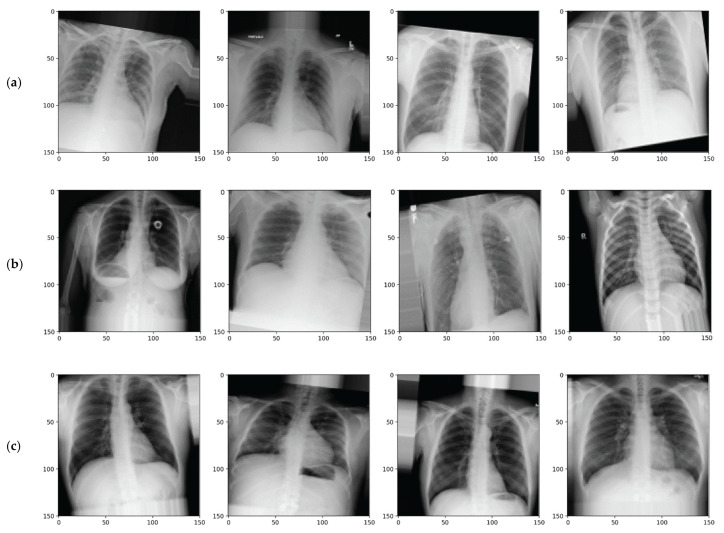
Example of pre-processed X-ray images for (**a**) COVID-19, (**b**) Normal, and (**c**) Viral Pneumonia.

**Figure 2 pathogens-12-00017-f002:**
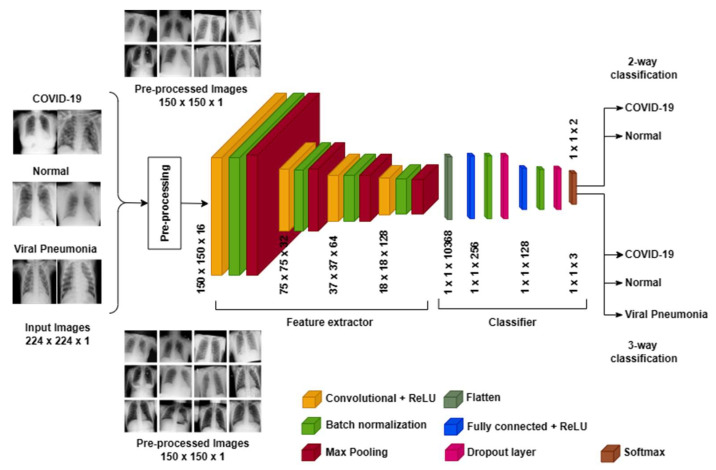
Proposed BNCNN model for COVID-19 detection.

**Figure 3 pathogens-12-00017-f003:**
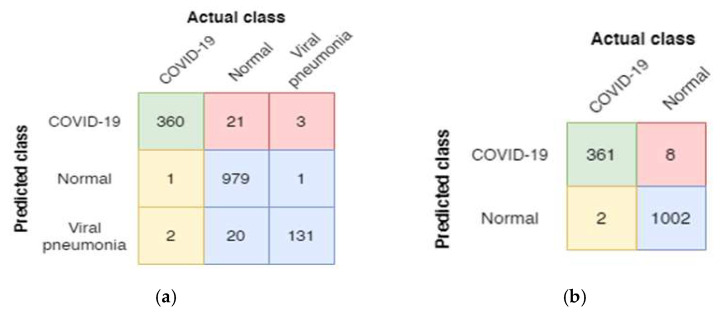
Confusion matrices of the BNCNN model for (**a**) three-way and (**b**) two-way classification (red: FP, yellow: FN, green: TP, and blue: TN for COVID-19 class).

**Figure 4 pathogens-12-00017-f004:**
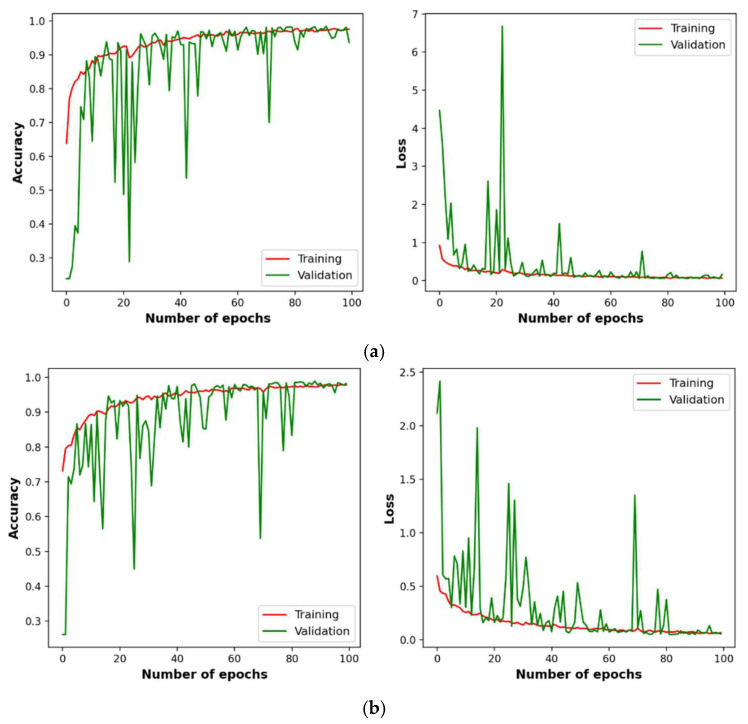
Variation of accuracy and loss of the BNCNN model on training and validation datasets over 100 epochs, (**a**) three-way and (**b**) two-way classification.

**Figure 5 pathogens-12-00017-f005:**
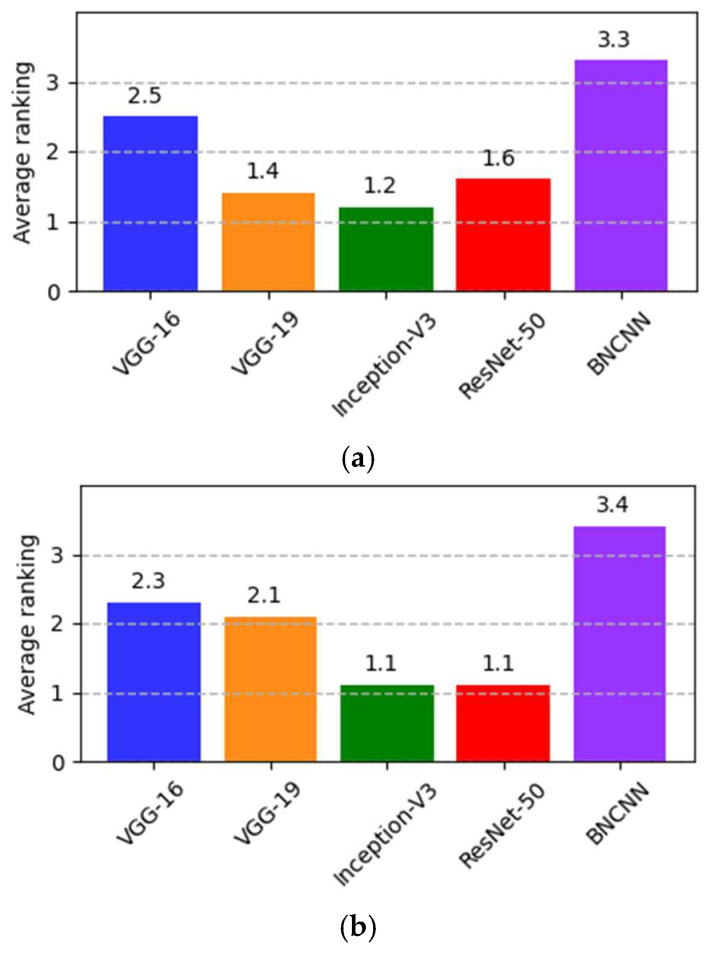
Average ranking based on testing accuracy at (the Friedman test) for (**a**) three-way (*p* = 0.0146) and (**b**) two-way (*p* = 0.0053) classification.

**Figure 6 pathogens-12-00017-f006:**
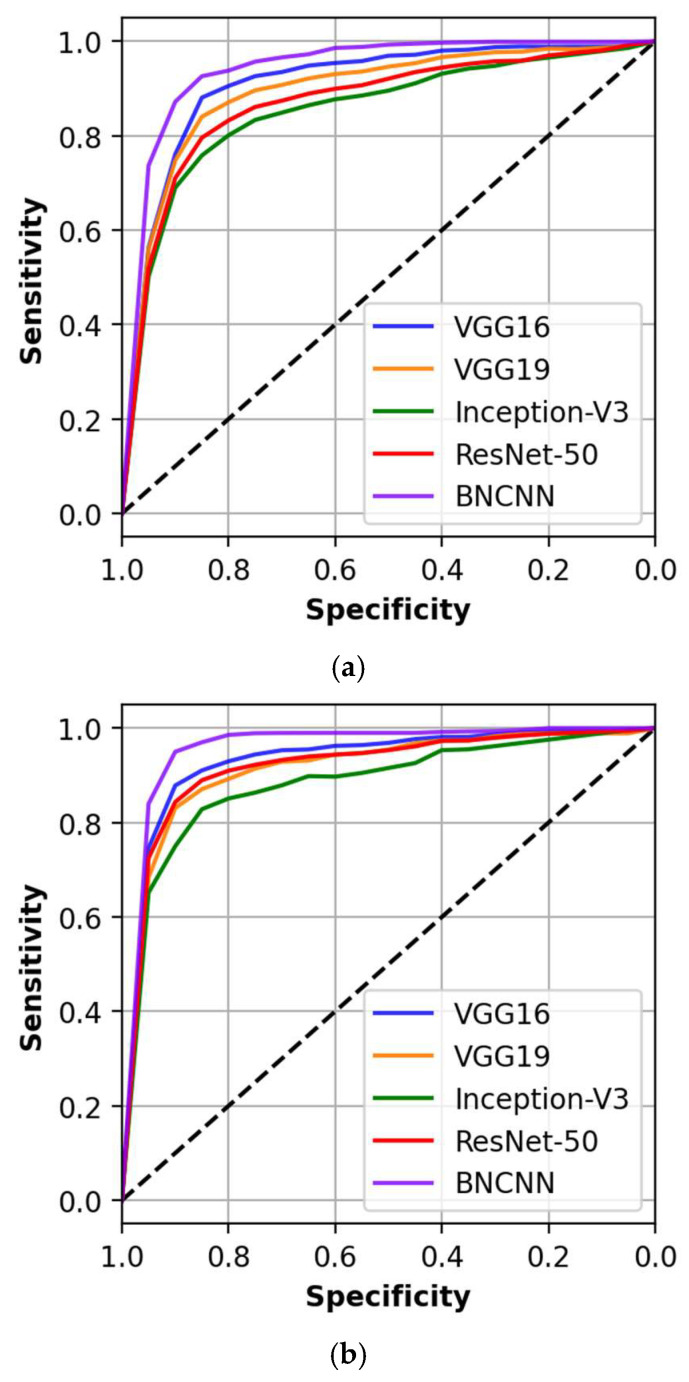
AUC and DeLong test for proposed BNCNN and other models for (**a**) three-way and (**b**) two-way classification.

**Table 1 pathogens-12-00017-t001:** Dataset description.

Dataset	COVID-19	Normal	Viral Pneumonia	Subtotal
Training	2892	8153	1076	12,121
Validation	362	1019	134	1514
Testing	362	1020	135	1518
Subtotal	3616	10,192	1345	15,153

**Table 2 pathogens-12-00017-t002:** A summary of the parameters used in the proposed BNCNN model.

Layer Type	Output Shape	Parameter Type	No. of Parameters
Convolution 2D	(None, 150, 150, 16)	Trainable	448
Batch Normalization	(None, 150, 150, 16)	Trainable + Non-trainable	32 + 32
Max. Pooling 2D	(None, 75, 75, 16)		0
Convolution 2D	(None, 75, 75, 32)	Trainable	4640
Batch Normalization	(None, 75, 75, 32)	Trainable + Non-trainable	64 + 64
Max. Pooling 2D	(None, 37, 37, 32)		0
Convolution 2D	(None, 37, 37, 64)	Trainable	18,496
Batch Normalization	(None, 37, 37, 64)	Trainable + Non-trainable	128 + 128
Max. Pooling 2D	(None, 37, 37, 64)		0
Convolution 2D	(None, 18, 18, 128)	Trainable	73,856
Batch Normalization	(None, 18, 18, 128)	Trainable + Non-trainable	256 + 256
Max. Pooling 2D	(None, 9, 9, 128)		0
Flatten	(None, 10368)		0
Dense	(None, 256)	Trainable	2,654,464
Batch Normalization	(None, 256)	Trainable + Non-trainable	512 + 512
Dropout	(None, 256)		0
Dense	(None, 128)	Trainable	32,896
Batch Normalization	(None, 128)	Trainable + Non-trainable	256 + 256
Dropout	(None, 128)		0
Softmax	(None, 3)/(None, 2)	Trainable	387/258
Total trainable			2,786,435/2,786,306
Total non-trainable			1248
Total			2,787,683/2,787,554

**Table 3 pathogens-12-00017-t003:** Hyper-parameter settings of Adam optimizer for different models.

Model	Learning Rate	Batch Size	Epochs
VGG-16	0.0001	16	30
VGG-19	0.0001	16	30
Inception-V3	0.001	8	30
ResNet-50	0.001	8	30
BNCNN	0.0001	16	30

**Table 4 pathogens-12-00017-t004:** Evaluation based on three-way class classification.

Model (Trainable/Total Parameters)	Dataset	Accuracy	Sen	PPV	F1
VGG-16 (165,379/14,880,835)	Training	96.68	96.81	96.88	96.84
	Validation	97.33	97.30	97.31	97.30
	Testing	96.64	93.01	96.81	94.87
VGG-19 (264,195/20,288,579)	Training	84.19	82.94	84.86	83.89
	Validation	89.20	88.92	89.68	89.30
	Testing	88.34	85.37	89.12	87.21
Inception-V3 (427,523/54,765,027)	Training	87.33	87.50	87.03	87.26
	Validation	91.76	92.24	91.19	91.71
	Testing	89.60	89.94	89.46	89.70
ResNet-50 (13,141,507/36,729,987)	Training	87.11	85.07	84.02	84.54
	Validation	87.78	86.36	88.37	87.35
	Testing	85.70	83.60	86.38	84.97
Proposed BNCNN (2,786,435/2,787,683)	Training	96.32	96.30	96.52	96.41
	Validation	97.49	97.43	97.48	97.45
	Testing	96.84	93.06	97.40	95.18

**Table 5 pathogens-12-00017-t005:** Evaluation based on two-way class classification.

Model (Trainable/Total Parameters)	Phase	Acc	Sen	PPV	F1
VGG-16 (165,250/14,880,706)	Training	96.61	96.61	96.63	96.62
	Validation	98.01	98.01	98.01	98.01
	Testing	94.28	94.29	94.29	94.29
VGG-19 (263,682/20,288,066)	Training	89.54	89.43	89.54	89.48
	Validation	92.61	92.62	92.61	92.61
	Testing	92.56	92.54	92.55	92.54
Inception-V3 (427,394/54,764,898)	Training	87.33	87.50	87.03	87.26
	Validation	91.76	92.24	91.19	91.72
	Testing	89.60	89.94	89.46	89.70
ResNet-50 (13,141,378/36,729,858)	Training	84.75	84.75	84.88	84.81
	Validation	89.20	89.34	89.12	89.23
	Testing	89.08	89.18	88.78	88.98
Proposed BNCNN (2,786,306/2,787,554)	Training	97.28	97.28	97.23	97.25
	Validation	98.55	98.52	98.53	98.52
	Testing	99.27	99.45	98.83	99.14

**Table 6 pathogens-12-00017-t006:** Accuracy comparison between the existing models and the proposed BNCNN model.

Model	3-Way Classification	2-Way Classification
Ozturk et al. (2020) [21]	87	98
Khan et al. (2020) [22]	95	N/A
Apostolopoulos and Mpesiana (2020) [23]	93.48	N/A
Wang et al. (2020) [24]	92.4	N/A
Sethy and Behera (2020) [25]	95.38	N/A
Horry et al. (2020) [26]	86	N/A
Rahimzadeh and Attar (2020) [27]	94.2	99.1
Song et al. (2021) [28]	81	N/A
Hussain et al. (2021) [29]	91.4	N/A
Chen (2021) [30]	85	N/A
Vinod et al. (2021) [31]	96.8	N/A
Anter et al. (2021) [32]	N/A	96
Basha et al. (2021) [33]	N/A	98.7
Proposed BNCNN model	**96.84**	**99.27**

**Table 7 pathogens-12-00017-t007:** Holm’s test for comparing BNCNN with other pre-trained models based on the testing accuracy.

3-Way Classification					2-Way Classification			
Competing Model	*p*-Value	Corrected *p*-Value	Null Hypothesis		Competing Model	*p*-Value	Corrected *p*-Value	Null Hypothesis
VGG-16	0.0956	0.5735	Reject		VGG-16	0.0597	0.2731	Reject
VGG-19	0.1165	0.5825	Reject		VGG-19	0.0049	0.0489	Reject
Inception-V3	0.0011	0.0105	Reject		Inception-V3	0.0232	0.1852	Reject
ResNet-50	0.0044	0.0397	Reject		ResNet-50	0.0186	0.1677	Reject

**Table 8 pathogens-12-00017-t008:** Sensitivity analysis for different models using the DeLong test.

	Competing Model	AUC	95% CI	*p*-Value
3-way classification	VGG-16	0.89	[0.85–0.92]	0.0412
	VGG-19	0.87	[0.82–0.93]	0.0431
	Inception-V3	0.84	[0.78–0.89]	0.0216
	ResNet-50	0.85	[0.79–0.90]	0.0312
2-way classification	VGG-16	0.90	[0.84–0.95]	0.0274
	VGG-19	0.87	[0.81–0.93]	0.0296
	Inception-V3	0.85	[0.80–0.89]	0.0012
	ResNet-50	0.88	[0.82–0.92]	0.0132

## Data Availability

The data presented in this study are openly available in Kaggle at [15].

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
