# Peer review of "A Deep Batch Normalized Convolution Approach for Improving COVID-19 Detection from Chest X-ray Images"

_pathogens, 2022, doi:10.3390/pathogens12010017_

Round 1

Reviewer 1 Report

1. The objectives of the paper should clearly explained.  The importance of the research are not included in abstract. 

2. Could be better if authors provide the augmentation process and how it is used for generation of images

3. Illustrate mathematical model for identification of max pooling?

4. What is the significance of this algorithm and who are the beneficiaries?

5. It would be nice if the author can provide the more description related to the dataset used

6. The conclusions need to show more consistency with the evidence and arguments presented.

7. Mathematical supporting equations are to be included. Results should be explain in detail about how comparison has done.

8. How the preprocessing has implemented?. Evidence has not provided.

9. Introduction and literature sections are too short and not meaningful.

Author Response

Response to Reviewers comments:

We are very much thankful to the reviewers for their deep and thorough review. We have revised our present research paper in the light of their useful suggestions and comments. We hope our revision has improved the paper to a level of their satisfaction.  Number wise answers to their specific comments/suggestions/queries are as follows.

Reviewer 1

  1. The objectives of the paper should clearly explained. The importance of the research are not included in abstract.

Authors' response: Thank you for your comment, research objectives have been modified

  1. Could be better if authors provide the augmentation process and how it is used for generation of images

Authors' response: Thank you for recommendation, Data augmentation technique is performed to enhance the quantity and variety of images given to the classifier for classification. Image augmentations used include zooming range, shearing, horizontal/ vertical stride. Etc. However, data augmentation along with parameters are provided in the manuscript.

"All subsets are augmented independently with a rotation range of [-10–10] degrees, zooming range of [0–10%], shearing of [0–10%], a horizontal stride of [0–10%], a vertical stride of [0–10%], and horizontally flip to improve generalization and increase diversity in the learning process by the models"

  1. Illustrate mathematical model for identification of max pooling?

Authors' response: Thank you for recommendation, a mathematical model of max pooling is added as recommended 

  1. What is the significance of this algorithm and who are the beneficiaries?

Authors' response: Thank a lot for raising this point, as it is well known, Coronavirus infected patients display very similar symptoms like pneumonia, and it attacks the respiratory organs of the body, causing difficulty in breathing. The disease is diagnosed using a Real-Time Reverse Transcriptase Polymerase Chain reaction (RT-PCR) kit and re-quires time in the laboratory to confirm whether the virus is exist or not. Due to insufficient availability of the kits in that time, the suspected patients cannot be treated in time, which in turn increases the chance of spreading the disease. To overcome this solution, radiologists observed the changes appearing in the radiological images such as X-ray. Going back to our proposed model, it is designed as a diagnostic system for Coronavirus in X-ray images, however, the need for deep learning with radiologic images is necessary for critical condition as this will provide a second opinion to the radiologists and healthcare community.

  1. It would be nice if the author can provide the more description related to the dataset used

Authors' response: Actually we provide all the available information about the dataset and the dataset can be access using the following link

https://www.kaggle.com/datasets/tawsifurrahman/covid19-radiography-database  

  1. The conclusions need to show more consistency with the evidence and arguments presented.

Authors' response: Conclusion section is improved as suggested

  1. Mathematical supporting equations are to be included. Results should be explain in detail about how comparison has done.

Authors' response: We have carefully reviewed if there is any mathematical supporting can be added as recommended and they are added where they needed. Results also improved based on reviewer 2 suggestions by adding 95% confidence intervals, AUC and DeLong test

  1. How the preprocessing has implemented?. Evidence has not provided.

Authors' response: Thank you for your comments, pre-processing process along with evidence is provided

  1. Introduction and literature sections are too short and not meaningful.

Authors' response: These sections are improved and changed as recommended by the second reviewer

Reviewer 2 Report

In this study, the authors have developed a novel batch normalized convolutional neural network (BNCNN) model to identify COVID-19 from chest X-ray images. The authors found that the proposed BNCNN model shows better performances than four other pre-trained models (VGG-16, VGG-19, Inception-V3 and ResNet-50) for 3-way and  2-way classification when applied to an open-access dataset.

 The paper is rather well organized and reads well.

 I have the following important comments

In general, accuracy must be reported with corresponding proportions used to calculate accuracies and with the corresponding 95% confidence intervals

As a general comment and to make the paper more readable for a larger community, I strongly suggest using Sensitivity instead of Recall in the whole manuscript. In addition, sensitivity must be reported with corresponding proportions used to calculate sensitivities and with the corresponding 95% confidence intervals (CI). 

Precision (wrongly written "percision" ) should be replaced by positive predictive value (PPV) and reported with corresponding proportions used to calculate PPV and with the corresponding 95% CIs.

Finally the authors must report specificities of each model with corresponding proportions and 95% CIs.

One important metric that is missing is the AUC of each model. This is an important issue for further comparison. In addition, the authors must compare each model in terms of AUC using the de Long test. It is important to know that when a model better performs, the difference in performances is significant or not. This is of paramount importance.

The authors must address the current limitations of chest X-ray for the diagnosis of Covid-19

As an example, in this study (Wong H  et al. Frequency and distribution of chest radiographic findings in patients positive for COVID-19 Radiology 2020; 296:E72-E7)8 one third of patients had negative chest X-ray manifestations on the first chest X-ray. This should be acknowledged.

 The paper lacks a Discussion section and a study limitation section as it should be. The section "Literature Review" should be titled Discussion and should be placed after the Results section. The following references must be added to place the study in a clinical perspective. This is a major limitation of the paper and this should be properly discussed with the following references.

Li J, Long X, Wang X, Fang F, Lv X, Zhang D, Sun Y, Hu S, Lin Z, Xiong N. Radiology indispensable for tracking COVID-19. Diagn Interv Imaging 2021;102(2):69-75.

Kato S, Ishiwata Y, Aoki R, Iwasawa T, Hagiwara E, Ogura T, Utsunomiya D. Imaging of COVID-19: An update of current evidences. Diagn Interv Imaging 2021;102(9):493-500.

Finally, the authors must discuss the limitations of AI models that use chest X-ray compare to those that use computed tomography .

Ito R, Iwano S, Naganawa S. A review on the use of artificial intelligence for medical imaging of the lungs of patients with coronavirus disease 2019. Diagn Interv Radiol 2020;26:443–8

Li L, Qin L, Xu Z, Yin Y, Wang X, Kong B, et al. Using artificial intelligence to

detect COVID-19 and community-acquired pneumonia based on pulmonary

CT: evaluation of the diagnostic accuracy. Radiology 2020;296:E65–71.

Author Response

We are very much thankful to the reviewers for their deep and thorough review. We have revised our present research paper in the light of their useful suggestions and comments. We hope our revision has improved the paper to a level of their satisfaction.  Number wise answers to their specific comments/suggestions/queries are as follows.

Reviewer 2

In this study, the authors have developed a novel batch normalized convolutional neural network (BNCNN) model to identify COVID-19 from chest X-ray images. The authors found that the proposed BNCNN model shows better performances than four other pre-trained models (VGG-16, VGG-19, Inception-V3 and ResNet-50) for 3-way and  2-way classification when applied to an open-access dataset.

 The paper is rather well organized and reads well.

Authors' response: Thanks a lot for your opinion in our work

I have the following important comments

In general, accuracy must be reported with corresponding proportions used to calculate accuracies and with the corresponding 95% confidence intervals

Authors' response:

As a general comment and to make the paper more readable for a larger community, I strongly suggest using Sensitivity instead of Recall in the whole manuscript. In addition, sensitivity must be reported with corresponding proportions used to calculate sensitivities and with the corresponding 95% confidence intervals (CI).

Authors' response: Sensitivity is used instead of Recall as suggested with CI as suggested

Precision (wrongly written "percision" ) should be replaced by positive predictive value (PPV) and reported with corresponding proportions used to calculate PPV and with the corresponding 95% CIs.

Authors' response: Precision replaced by positive predictive value (PPV) with CI

Finally the authors must report specificities of each model with corresponding proportions and 95% CIs.

Authors' response: Thank you for comment, specifications for each used model are provided in Table 3-.section experimental set-up

One important metric that is missing is the AUC of each model. This is an important issue for further comparison. In addition, the authors must compare each model in terms of AUC using the de Long test. It is important to know that when a model better performs, the difference in performances is significant or not. This is of paramount importance.

Authors' response: thanks a lot for your suggestion, each model in terms of AUC using the de Long test is provided

The authors must address the current limitations of chest X-ray for the diagnosis of Covid-19, As an example, in this study (Wong H  et al. Frequency and distribution of chest radiographic findings in patients positive for COVID-19 Radiology 2020; 296:E72-E7)8 one third of patients had negative chest X-ray manifestations on the first chest X-ray. This should be acknowledged.

Authors' response: limitations of chest X-ray are discussed as suggested and highlighted using provided reference

 The paper lacks a Discussion section and a study limitation section as it should be. The section "Literature Review" should be titled Discussion and should be placed after the Results section. The following references must be added to place the study in a clinical perspective. This is a major limitation of the paper and this should be properly discussed with the following references.

Li J, Long X, Wang X, Fang F, Lv X, Zhang D, Sun Y, Hu S, Lin Z, Xiong N. Radiology indispensable for tracking COVID-19. Diagn Interv Imaging 2021;102(2):69-75.

Kato S, Ishiwata Y, Aoki R, Iwasawa T, Hagiwara E, Ogura T, Utsunomiya D. Imaging of COVID-19: An update of current evidences. Diagn Interv Imaging 2021;102(9):493-500.

Authors' response: Thank you for your comment, literature review section content has been shifted comparative with existing models subsection and a discussion section is added as recommended. In the discussion section, we discuss the results and focus the light on the limitations of our study to make the work flow well. For the limitations, provided references are used as suggested.   

Finally, the authors must discuss the limitations of AI models that use chest X-ray compare to those that use computed tomography .

Ito R, Iwano S, Naganawa S. A review on the use of artificial intelligence for medical imaging of the lungs of patients with coronavirus disease 2019. Diagn Interv Radiol 2020;26:443–8

Li L, Qin L, Xu Z, Yin Y, Wang X, Kong B, et al. Using artificial intelligence to

detect COVID-19 and community-acquired pneumonia based on pulmonary

CT: evaluation of the diagnostic accuracy. Radiology 2020;296:E65–71.

Authors' response: AI models that use chest X-ray compare to those that use computed tomography have been discussed and added in discussion section 

Round 2

Reviewer 1 Report

I am not satisfied with the revised paper. I don't think the authors revised the paper according to my recommendation. 

1. The objective is not stated clearly in this research. I raised this same query in first round of review. 

2. Introduction and literature review section contents are not covering basic knowledge of research. I recommended to include the additional information but this revised paper, removed more contents. 

3. Mathematical equations are not included but in author response made the answer as made it clear. 

4. Preprocessing evidence also not given. 

5. Overall, this paper not revised and improved quality. 

Author Response

Thank you for allowing a resubmission of our manuscript, with an opportunity to address the reviewers’ comments.

Editor Notes

Dear Authors, the objective is not stated clearly in this research.

Guest Editors suggest to clarify it in the abstract section, as well as in the final part of the Introduction.

More over Guest Editors suggest to revise the parts of the paper claiming the technical novelty of the proposed network, as it represents a dear and old Convolutional Neural Network with few layers. Nowadays, a lot of advances have been proposed by specialized literature.

Ans: Thank you for the review. The paper's primary objective is to build a CNN-based generalized feature extractor with minimum computational complexity, which should be usable in the binary and multi-class classification of COVID-19 using chest X-ray images. Although the proposed BNCNN uses state-of-the-art layers, their arrangement and hyper-parameters are carefully selected through various literature references and empirical investigations. The most recent networks may have comparable performance to the proposed BNCNN model but at the cost of much higher computational cost than the proposed BNCNN model.

Reviewer 1

I am not satisfied with the revised paper. I don't think the authors revised the paper according to my recommendation.

  1. The objective is not stated clearly in this research. I raised this same query in first round of review.

Ans: Thank you for the review. We have revised the abstract and contribution sections of the paper to emphasize the research objective more clearly.

  1. Introduction and literature review section contents are not covering basic knowledge of research. I recommended to include the additional information but this revised paper, removed more contents.

Ans: Thank you for the review. We have revised the introduction section to emphasize the research development. The primary research knowledge is retained in Section 1, 'Introduction', while the additional paper in the literature is shifted to Section 4.3, 'Comparison with existing models', for better correlation with results and discussion.

  1. Mathematical equations are not included but in author response made the answer as made it clear.

Ans: Thank you for the review. We have added Eq. (1) for feature extraction and Eq. (2) for loss calculation in the classifier. The proposed BNCNN network uses a state-of-the-art layer in a novel configuration. Hence, we have not claimed the new development of layers by adding mathematical equations for individual layers. The detailed mathematical analysis for all layers and kernel functions can be obtained in Ref. [16], as cited in the paper.

  1. Pre-processing evidence also not given.

Ans: Thank you for the review. Section 2.1. 'Data pre-processing' describes the steps involved in pre-processing. The parameter settings and reasoning for the choice of each step are mentioned in the section. Figure 1 shows example evidence of the pre-processed images for each class. The figure caption is revised for better understanding and correlation of the readers.

  1. Overall, this paper not revised and improved quality.

Ans: Thank you for the review. The paper is revised to enhance the objective, contribution, and literature sections. Also, grammatical errors are eliminated using Professional English software and proofreading by a native English reader.

Round 3

Reviewer 1 Report

Present form can be consider